# A fungal phospholipase C involved in the degradation of plant glycosylinositol phosphorylceramides during Arabidopsis/Botrytis interaction

Luka Lelas[1], Justine Rouffet[1,2], Alexis Filachet [1], Julien Sechet[1,3], Antoine Davière[1], Thierry Desprez[1], Samantha Vernhettes [1] ✉ & Aline Voxeur [1] ✉

This study investigates the presence and significance of phosphorylated oligosaccharides that accumulate during the interaction between *Arabidopsis thaliana* and *Botrytis cinerea*, a necrotrophic fungus that poses a major threat to crops worldwide. While previous research has extensively characterized cell wall-derived molecules during fungal infection, the role of plasma membrane-derived ones remains unclear. Here, we reveal the discovery of inositol phosphate glycans (IPGs) released during infection, originating from plant sphingolipids, specifically glycosylinositol phosphorylceramides (GIPC). Advanced chromatography, mass spectrometry techniques and molecular biology were employed to identify these IPGs, and determine their origins. In addition to the well-characterized role of *B. cinerea* in releasing cell wall-degrading enzymes, this research suggests that *B. cinerea*'s enzymatic machinery may also target the degradation of the plant plasma membrane. As a consequence of this, IPGs identical to those generated by the host plant are released, most likely due to activity of a putative phospholipase C that acts on GIPC plasma membrane lipids. These insights could pave the way for developing new strategies to enhance crop resistance by focusing on membrane integrity in addition to cell wall fortification.

*Botrytis cinerea*, commonly known as the gray mold fungus, is a necrotrophic pathogen that can infect a wide range of plant species. This fungus is particularly known for its aggressive plant cell wall degradation, a process that raises critical questions about its biological mechanisms and the plant's defense responses[1]. Elucidating the intricacies of *B. cinerea*'s biology and its interactions with host plant enables the development of sustainable and effective strategies to mitigate its impact on agriculture, benefiting both the scientific community and society at large.

Oligosaccharins are biological active oligosaccharides that can be derived from plant cell walls and be produced upon plant pathogen interaction[2]. They are known for their diverse effects in plants, including promoting growth in low- and normal- light conditions[3–7] and anti-auxin activity[8–11]. Moreover, while we have known for decades that oligogalacturonides (OGs), α-1,4-linked galacturonic acid oligomers derived from pectins, trigger plant defense responses upon exogenous application[12–14], the

detection and structural characterization of these oligosaccharides within plants infected by *B. cinerea* have been recent developments[15]. In addition, *in muro* release of OGs has been achieved through expression of a fungal polygalacturonase gene (*PG*) fused with a gene encoding a plant polygalacturonase-inhibiting protein[16].

To achieve a comprehensive characterization of the oligosaccharides generated during the *A. thaliana/B. cinerea* interaction, beyond just pectin-derived oligosaccharides, we performed the structural characterization of the oligosaccharides accumulated over time. We have uncovered, to the best of our knowledge, previously unknown oligosaccharides that accumulate during the interaction between *Arabidopsis thaliana* and *B. cinerea*. In contrast to our initial expectations of characterizing cell wall-derived oligosaccharides, we have identified a substantial presence of inositol phosphate glycans (IPGs) which become highly abundant during infection and originate from vital and

[1]Université Paris-Saclay, INRAE, AgroParisTech, Institut Jean-Pierre Bourgin (IJPB), 78000 Versailles, France. [2]Present address: Institut Agro, Univ Angers, INRAE, IRHS, SFR QuaSaV, 49000 Angers, France. [3]Present address: AlkInnov, Innovation for Life, 92100 Boulogne-Billancourt, France. ✉e-mail: samantha.vernhettes@inrae.fr; aline.voxeur@inrae.fr

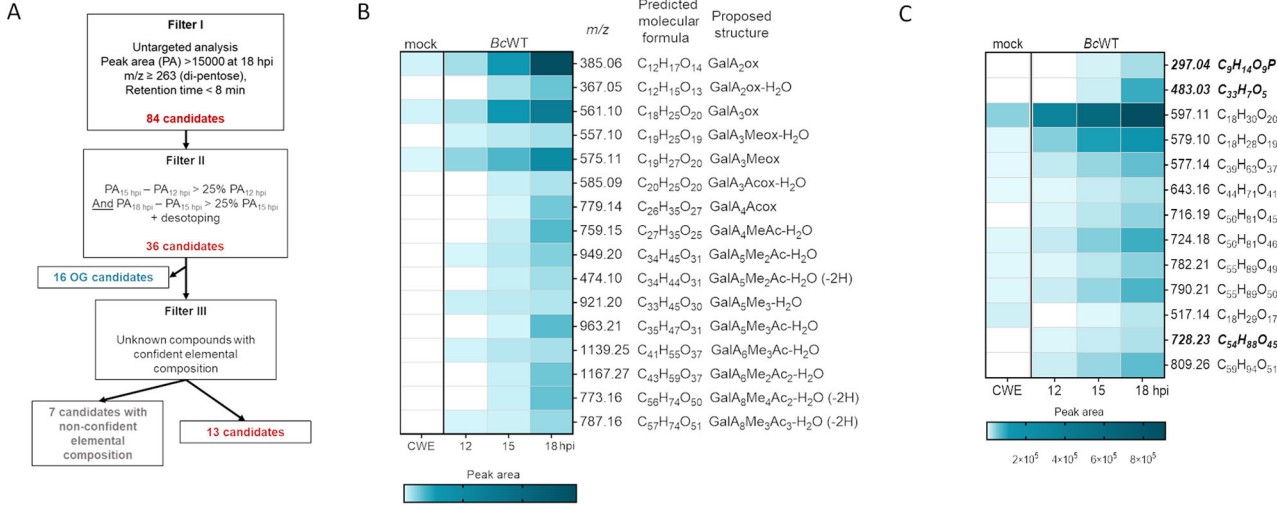

**Fig. 1 | Candidate selection using a three-step filtering strategy. A** Flowchart illustrating the three-step filtering strategy employed to select potential oligosaccharides that accumulate during infection. **B** Heat map displaying the putative oligogalacturonides (OGs) accumulating over time during the infection. **C** Heat map showing the non-OG candidates accumulating over time during the infection. Candidates in bold and italic lack elemental compositions compatible with an

oligosaccharide structure. PA Peak area, hpi hours post infection, OG oligogalacturonides, CWE cell wall water extract, BcWT *B. cinerea* wild type strain, GalA galacturonic acid, Me methylester group, Ac acetylester group; OGs are named GalA$_x$Me$_y$Ac$_z$. Subscript numbers indicate degree of polymerization and the number of methyl- and acetylester groups, respectively.

intricate lipids of plant cell membranes: the glycosylinositol phosphorylceramides (GIPCs).

## Results

### Unknown oligosaccharides accumulate during *A. thaliana/B. cinerea* interaction

To begin, we employed high-performance size-elusion chromatography (HP-SEC) in conjunction with a high-resolution mass spectrometry (HRMS) method in negative mode to establish a systematic approach for identifying potential oligosaccharides that accumulate during infection over time. We focused on ions consistently increasing between 12- and 18-h post-infection, possessing a mass-to-charge ratio ($m/z$) exceeding 263 and a retention time below 8 min, indicative of a dipentose (Fig. 1A, Dataset S1). This analysis led to the identification of 36 candidates, which included 16 candidates with predicted formulae and retention time compatible with OGs we have previously identified[15], four of which are likely oxidized through activity of plant OG-oxidizing enzymes from the berberine bridge enzyme-like family[17] while 12 were unsaturated (-H$_2$O) and likely result from pectin lyase activity[15] (Fig. 1B). Out of the remaining 20 candidates, seven were excluded due to the unreliable elemental compositions. Consequently, our methodology led to the recognition of 13 candidates, with nine being phosphorylated, while three lacked elemental compositions compatible with an oligosaccharide structure (Fig. 1C). Remarkably, eight of these candidates were also detected in the cell wall water extract of non-infected plants.

### Acetylated xyloglucan-derived oligosaccharides and inositol phosphate glycans are produced during *A. thaliana/B. cinerea* interaction

The candidate with the largest molecular formula at $m/z$ 809 is doubly charged (predicted formula C$_{60}$H$_{98}$O$_{50}$) and was detected as containing a formic acid adduct $[M + HCOO^- - H]^{2-}$. Its fragmentation resulted in the formation of its $[M-H]^-$ form at $m/z$ 1573, along with ions at $m/z$ 161 (anhydro-hexosyl unit), 1411 [M-C$_6$H$_{10}$O$_5$-H]$^-$ and 1279 [M- C$_6$H$_{10}$O$_5$- C$_5$H$_8$O$_4$-H]$^-$ suggesting the presence of hexosyl and pentosyl residues (Fig. S1). We attributed the doubly charged ion at $m/z$ 726, produced via a 120-Da loss, to a $^{2,4}$A$_n$ cross-ring cleavage in a C$_4$-substituted hexose. Ions at 675 [M-Hex-C$_2$H$_4$O$_2$-2H]$^{2-}$ and 666 [M-Hex-C$_2$H$_4$O$_2$-H$_2$O-2H]$^{2-}$ were assigned to a $^{0,2}$A$_{n-1}$ cross-ring cleavage which is diagnostic of C$_4$- and C$_6$-

substituted hexose. The prominent and smaller single charged fragments at $m/z$ 643 and 455 were accompanied by ions at $m/z$ 541 and 353 respectively, corresponding to a 102-Da neutral loss. This MS$^2$ pattern is characteristic of xyloglucan oligosaccharide fragmentations known for their prominent D-type fragment ions. These fragments result from double cleavage events and correspond to entire inner xyloglucan side chains[18]. They produce $^{0,4}$A$_i$ cross-ring cleavage ions, diagnostic for the presence of (1,6)-linkage, via a 102-Da loss from anhydro-glycosyl unit. Consequently, we attributed the D-type ion at $m/z$ 455 and its $^{0,4}$A$_\beta$ corresponding ion at $m/z$ 353 to the presence of Galactose-Xylose-Glucose block (represented by the letter L). Additionally, $m/z$ 643 (D), 541 ($^{0,4}$A$_{i\alpha}$) and 499 ($^{0,4}$A$_{i\alpha}$ -Acetyl) were attributed to the presence of an acetylated Fucose-Galactose-Xylose-Glucose block (represented by the letter F). Based on these results and the well-known structure of xyloglucan, we concluded that the candidate at $m/z$ 809 is an acetylated XLFG oligosaccharide.

We next focused our study on the most abundant candidate at m/z 597 $[M - H]^-$, with the software-predicted formula C$_{18}$H$_{30}$O$_{20}$P$^-$. This compound coelutes with the m/z 579 ion (predicted formula C$_{18}$H$_{29}$O$_{19}$P$^-$) which we attributed to the formation of a [M-H-H2O] anion. The $m/z$ 597 MS$^2$ spectrum displays a $m/z$ 97 ion corresponding to H$_2$PO$_4^-$, conforming to the predicted formula (Fig. 2A). Furthermore, the $m/z$ 259 and 241 ions are diagnostic of phosphorylated inositol (Ino-P), being notably found in MALDI-MS/MS analysis of GIPCs[19–22]. The ion at $m/z$ 241 was next assigned the loss of a uronic acid (UA) from the $m/z$ 373 ion, shown to represent a useful signature of the GIPC polar head[19,20]. It is likely formed from the decarboxylation of the $m/z$ 417 ion, the latter corresponding to the loss of a hexose (Hex) moiety from the parental ion $[M-H-180]^-$. Altogether, this fragmentation pattern suggests that $m/z$ 597 ion corresponds to an inositolphosphate glycan (IPGs) composed of a phosphorylated inositol, a uronic acid and a hexose that might originate from *A. thaliana* Series A GIPCs. We next fragmented the candidate at $m/z$ 517 (C$_{18}$H$_{29}$O$_{17}^-$) (Fig. 2B). We observed fragment ions at $m/z$ 161 and 179 corresponding to hexosyl or inositol residue. We also detected characteristic fragment ions of uronic acids at m/z 175 [M-H$_2$O-H]$^-$ and 113 [M-COO]$^-$. Being consistent with the $m/z$ 597 fragmentation mechanism, the fragment ion at $m/z$ 293 is formed by the loss of a hexose or an inositol and a further decarboxylation. We conclude that $m/z$ 517 likely corresponds to the non-phosphorylated form of the $m/z$ 597 candidate.

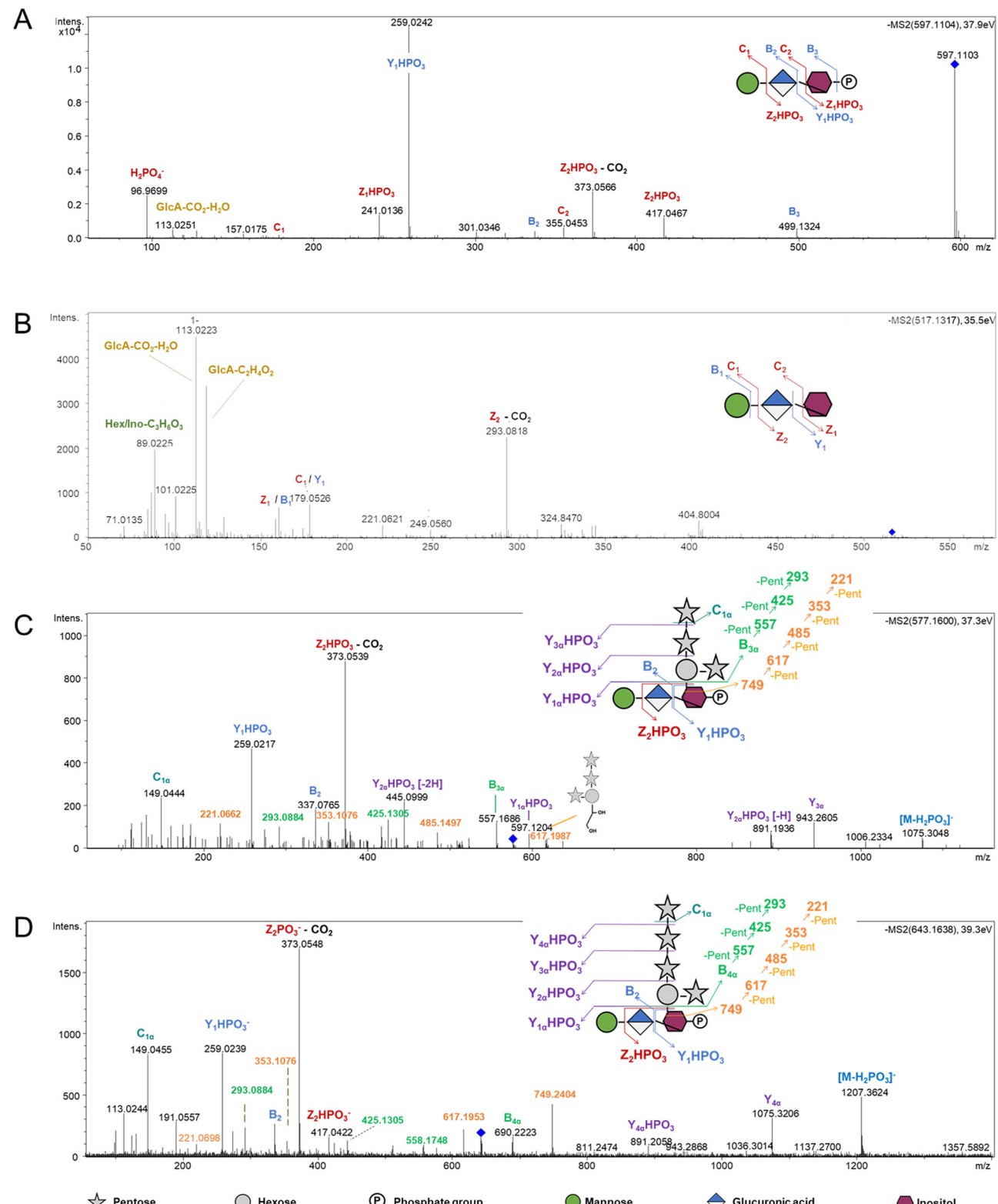

**Fig. 2 | Inositol (phosphate) glycans (I(P)Gs) accumulate during *Arabidopsis thaliana-Botrytis cinerea* infection. A** MS² fragmentation pattern of *m/z* 597 in negative mode. **B** MS² fragmentation pattern of m/z 517 in negative mode. **C** MS² fragmentation pattern of m/z 577 in negative mode. **D** MS² fragmentation pattern of m/z 643 in negative mode. GlcA: Glucuronic acid, Intens.: signal intensity. Pent: Pentose.

The *m/z* 259 (Ino-P) and 373 (UA-Ino-P) ions observed in *m/z* 597 MS² spectrum were found as well in the fragmentation pattern of six doubly charged candidates at m/z 577 ($C_{39}H_{63}O_{37}P^{2-}$), 643 ($C_{44}H_{71}O_{41}P^{2-}$), 716 ($C_{50}H_{81}O_{45}P^{2-}$), 724 ($C_{50}H_{81}O_{46}P^{2-}$), 782 ($C_{55}H_{89}O_{49}P^{2-}$), and 790 ($C_{55}H_{89}O_{50}P^{2-}$). According to the fragmentation pattern obtained from *m/z* 577 and 643 (Fig. 2C, D), these molecules contain one additional hexose and three or four additional pentoses respectively, likely linked to a Hex-UA-Ino-P core.

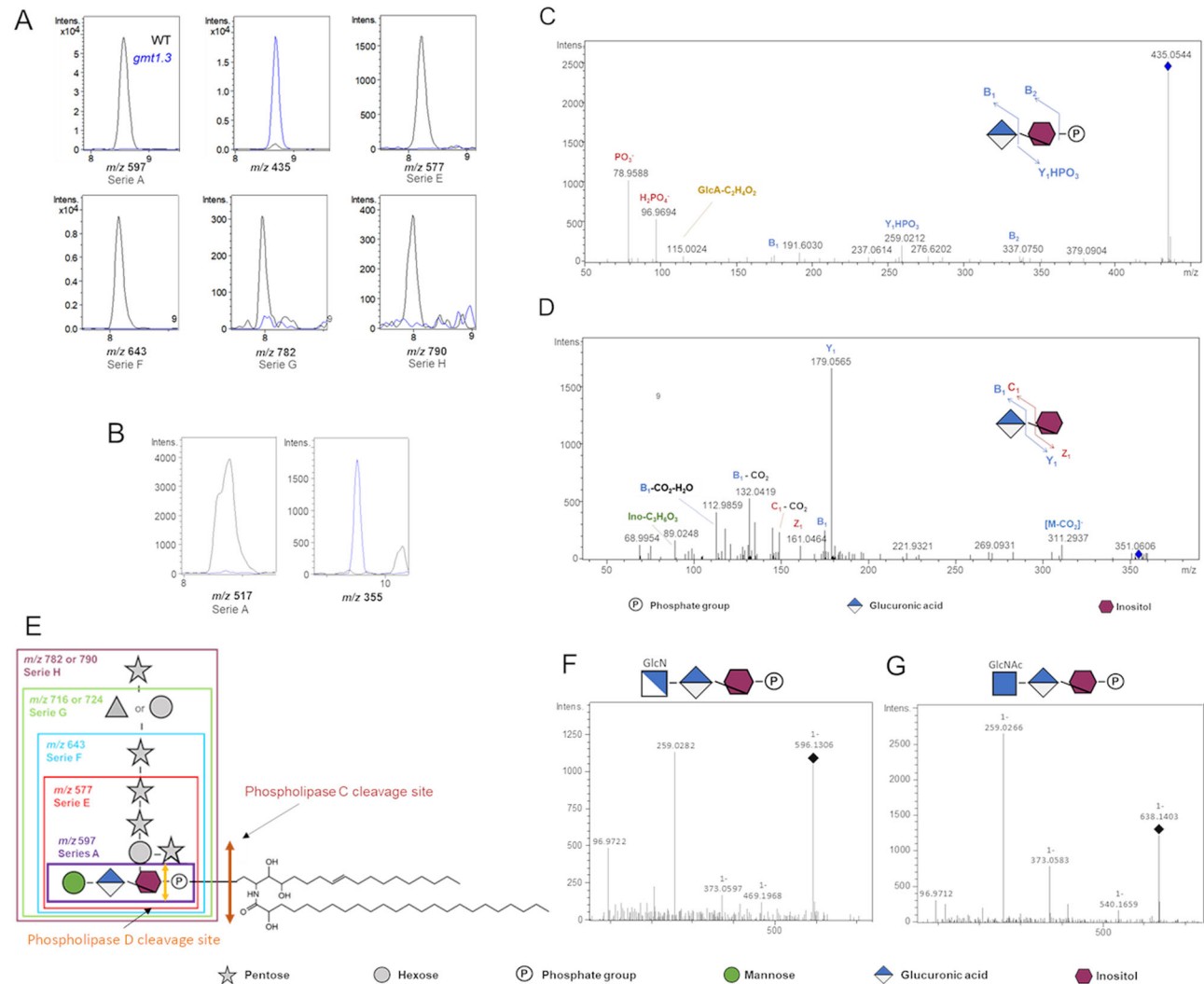

**Fig. 3 | Inositol (Phosphate) Glycans (I(P)Gs) derive from plant glycosylinositol phosphorylceramides (GIPCs).** Extracted ion chromatograms of IPG (**A**) and IG molecules (**B**) in cell wall water extract of wild-type and *gmt1.3* mutant *A. thaliana* 14-day-old plantlets. MS² fragmentation pattern of m/z (**C**) 435 and (**D**) 355, detected in *gmt1.3* mutant. **E** Structural composition of the different GIPC Series detected in *A. thaliana*. Fragmentation pattern of m/z (**F**) 596 and (**G**) m/z 608 detected in tomato leaves infected by *B. cinerea*. Intens.: signal intensity. GlcN(Ac): (N-acetyl)glucosamine.

The cross-ring fragments at m/z 353, 485, 617 and 749 ($B_{4\alpha} + C_2H_4O_2$) might also be produced by the cleavage of an inositol or a hexose residue. We favor the first hypothesis since no fragment corresponding to three or four pentoses plus two hexoses has been detected. Last, the cross-ring fragment at *m/z* 221 ($C_6H_{10}O_5 + C_2H_4O_2$) provides evidence that the Pent₄Hex chain and Pent₃Hex are branched on the inositol moiety via the hexose residue. We conclude that the *m/z* 577 and 643 candidates were IPGs.

Following the same reasoning, according to the fragmentation pattern obtained from candidates at *m/z* 716 and 724 (Fig. S2A, S2B), we attributed their respective fragments at *m/z* 895 and 911 to the presence of side chains composed of one hexose linked to the Hex-UA-Ino-P core along with four pentoses and either one additional deoxyhexose (*m/z* 716) or a hexose (*m/z* 724). We finally deduced that the candidates at *m/z* 782 and 790 are characterized by one additional pentose (Fig. S2C, S2D). Therefore, the candidates at *m/z* 716 and 724 on one hand, and *m/z* 782 and 790 on the other were also IPGs.

### Inositol phosphate glycans originate from plant sphingolipids

To explore whether the I(P)Gs mentioned above originate from the plant sphingolipids known as glycosyl inositol phosphoryl ceramides (GIPCs),

we analyzed the cell wall water extract of *gmt.1.3* (GIPC Mannosyl Transferase) mutant which is impaired in GIPC glycan biosynthesis[23]. We examined 14-day-old wild-type and *gmt1.3* *loss-of-function* mutant plants. Cell wall water extract of WT plantlets revealed six of the IPGs and an IG similar to ones detected upon infection (Fig. 3A, B). In contrast, we have not found any of the previously mentioned IPGs in the identical extract prepared from the *gmt1.3* mutant. Also, we observed ions at *m/z* 435 ($C_{12}H_{20}O_{12}P$) and 355 ($C_{12}H_{20}O_{12}$) that were absent in the WT samples (Fig. 3A, B) and probably correspond to GlcA-Ino-P and GlcA-Ino structures, respectively. The fragmentation of *m/z* 435-originating ions at *m/z* 259, is likely resulting from the loss of a uronic acid, as well as ions at *m/z* 97 and 79, which are indicative of dihydrogen phosphate and hydrogen phosphite groups (Fig. 3C). The fragmentation of m/z 355 suggested the presence of a $C_6H_{10}O_6$ moiety and a uronic acid not present in the WT samples, which we attributed to GlcA-Ino (Fig. 3D). It is worth to note that we did not detect any ion corresponding to the GlcA-Ino core substituted by side chains, raising questions of the underlying role of GIPC glycan biosynthetic machinery of the *gmt1.3* plants. In conclusions, we deduced that the seven IPGs and the IG that accumulated during *A. thaliana*/*B. cinerea* interaction derive from Series A and Series E to H GIPCs and are likely released by GIPC phospholipase(s) C and D that

cleave between the phosphate and the lipid moiety, and the phosphate and the glycan moiety, respectively (Fig. 3E).

To further confirm that IPGs accumulated upon infection originate from plant GIPC, we employed the same methodology but utilized tomato, infected by *B. cinerea*. The relevance of this lies in the fact that the polar heads of tomato GIPC contain an N-acetylglucosamine (GlcNAc) residue instead of mannose, a specificity of the *Solanaceae* family[24]. The resulting observations pointed to an ion at m/z 596 [M - H]-, for which the software-predicted formula is $C_{18}H_{31}O_{19}NP^-$ and another at 638 ($C_{20}H_{33}O_{20}NP^-$). The variance in molecular formula between these two ions corresponds to an acetyl group ($C_2H_2O$). Finally, upon fragmentation of these ions, we observed ions at m/z 259 and 373, the characteristic diagnostic ions originating from the glycan-based polar head of the GIPC molecule[16,17] (Fig. 3E–G). We conclude that the ions at m/z 596 and 638 correspond to Series A and acetylated Series A GIPC polar heads thus not only indicating IPGs are produced upon infection of other plant families but also revealing differences in their structural composition.

Owing to their structure and class of molecules they belong to, we hypothesize IPGs and IGs are likely released by GIPC-specific phospholipases C and D which cleave the bond before and after the phosphate, respectively (Fig. 3E).

### *B. cinerea*'s enzymatic machinery is able to generate inositol phosphate glycans

Given that GIPC-phospholipase activity has already been reported in *A. thaliana*, we explored the previously unreported ability of *B. cinerea* to degrade GIPCs. We first incubated *B. cinerea* with GIPC-enriched lipidic *A. thaliana* extract (which already contained a small amount of co-extracted IPGs, Fig. S3C) and as negative control, we incubated *B. cinerea* with commercial pectins which not contain GIPCs and on which *B. cinerea* is known to grow well[22]. In the presence of pectins and *B. cinerea* conidia, we detected the production of uridine diphosphate N-acetylglucosamine (UDP-GlcNAc; m/z 606) but no IPGs. Conversely, in the absence of the fungus, UDP-GlcNAc was not produced, clearly indicating that UDP-GlcNAc is a marker of *B. cinerea* growth (Fig. S3A, B). By contrast, in the GIPC-enriched lipidic extract, whether in the presence or absence of *B. cinerea* conidia, we were not able to detect either UDP-GlcNAc production or additional IPG production (Fig. S3C, D). Such an outcome strongly points to the inability of *B. cinerea* to grow on GIPC-enriched lipidic extract rather than bringing to question its inability to digest it enzymatically.

As an alternative, we prepared substrates from *A. thaliana* leaves using 96% ethanol at room temperature in order to produce an alcohol-insoluble residue (AIR) fraction which contains both the cell wall and plasma membrane, since GIPC and phospholipids extraction demands usage of heated ethanol[19]. We incubated *B. cinerea* conidia with this AIR, and monitored IPG and UDP-GlcNAc production over time. We detected UDP-GlcNAc within the first hours of incubation suggesting that the fungus grew well and its level remained stable over time. More interestingly, we observed the gradual accumulation of IPGs from Series A to H (Fig. 4A). This finding consolidates the notion that IPG production is indeed a result of degradation of plant GIPCs by one or several putative *B. cinerea* phospholipases.

Finally, to determine the effect of *B. cinerea* conidia on AIR GIPCs, we extracted them from inoculated and non-inoculated AIR using a solution of acidic, hot ethanol. The extracted liquid was left to precipitate in freezing conditions and the obtained pellet was resuspended in water before HPSEC-MS analysis. When analyzing the mock, we detected the presence of the Series A GIPCs composed of long-chain base t18:1 and fatty acid chains h22:0 to h26:0 (where t18:1 indicates a trihydroxylated long-chain base with 18 C atoms and one C = C bond, and h22:0 indicates a monohydroxylated fatty acid with 22 C atoms and no C = C bonds) and their corresponding phytoceramide (t18:1 h22:0; t18:1 h24:0 and t18:1 h26:0) (Fig. 4B, C). In comparison, in the reaction where conidia were included, both this GIPC and its corresponding phytoceramide are present, albeit the relative abundance of the phytoceramide being much higher compared to the mock infection. It is worth to note that we did not detect a clear decrease of Series A peak area that might be due to an unexpected matrix ionization effect. Accompanying the increase in relative abundance of the phytoceramide t18:1 h24:0, we detected the core IPG motif Man-GlcA-Ino-P abundantly, along with GalA₃ and GalA₄Me oligosaccharides originating from pectin degradation (Fig. 4C). Finally, we incubated BcWT conidia with AIR extracted with a solution of acidic, hot ethanol (GIPC-free AIR) and observed that far less Series A IPG were produced (Fig. S4). Altogether, these results clearly show abundant GIPC degradation occurs due to the enzymatic activity of *B. cinerea*.

### BCIN07g04350 is a Series A GIPC phospholipase C

To investigate *B. cinerea* GIPC phospholipases, we first selected putative sphingomyelinases, encoded by a family of three genes (BCIN03g00290, BCIN03g04310, and BCIN07g04350), as potential GIPC phospholipase candidates, given that sphingomyelin in animals is analogous to GIPCs in plants. Notably, orthologs were found in almost all well-known and economically significant fungal phytopathogens with various lifestyles (Fig. S5), except in all species of the Alternaria genus. Moreover, we found that the *B. cinerea*'s sphingomyelinases were divided into two clades: BCIN03g00290 and BCIN03g04310 in the upper clade, and BCIN07g04350 in the lower clade, suggesting that BCIN07g04350 might have a different substrate specificity than the other two.

According to Zhang et al.[25], both *BCIN07g04350* and *BCIN03g00290* expression at 16 hpi display significant correlations to 72 hpi lesion area across the three *A. thaliana* genotypes tested (Table S1). Furthermore, data from Souibgui et al.[26] showed that BCIN07g04350, in contrast to the two other putative sphingomyelinases, was secreted when conidia were grown in CCPX medium, which contains several polysaccharides such as carboxymethyl cellulose, polygalacturonic acid, and xylan, suggesting that BCIN07g04350 is expressed, on the contrary to the two others, when *B. cinerea* conidia is incubated with *A. thaliana* AIR.

We therefore focused on BCIN07g04350 as a putative BcGIPC-PLC1 and expressed it in the *E. coli*/*P. pastoris* heterologous expression system. Following confirmation of the protein in the culture medium (Fig. S6), we used it on AIR substrates prepared from *A. thaliana* leaves as described above. LC-MS analysis of these tests showed GIPC-PLC activity on Series A GIPCs while also revealing the protein's preference for an acidic pH. BCIN07g04350 also acted to a lesser extent on Series F, G and H GIPCs that might reflect the low content of AIR in highly glycosylated GIPCs (Fig. 5A) but also poses the question if *B. cinerea* might secrete other GIPC-PLCs with varying substrate specificities to degrade highly glycosylated GIPCs. We also monitored the production of polar heads of more common phospholipids and did not detect any production suggesting that BCIN07g04350 is GIPC specific (Fig. S7).

We then tested the specificity of the putative GIPC-PLC1 BCIN07g04350 by incubating it with GIPC-enriched lipid fraction prepared from *A. thaliana* leaves and analyzed the hydrolysate by HPEC-MS. In the mock, ions at *m/z* 1232.69, 1260.71 and 1288.75 corresponding to Series A t18:1 h22:0, Series A t18:1 h24:0 and Series A t18:1 h26:0 GIPCs were detectable in abundance as opposed to the also present core IPG motif with *m/z* 597 (Figs. 5B, C). On the other hand, when analyzing GIPC-enriched lipid fraction treated with BCIN07g04350, together with the presence of the same GIPCs, we were able to identify their corresponding phytoceramide parts at *m/z* 652.58, 678.60, 680.61 and 694.59, corresponding to phytoceramides t18:1 h22:0, t18:1 h24:1, t18:1 h24:0 and t18:1 h25:0. However, it is worth to note that only a few of t18:1 h26:0 was detected. Along with that, now we identified the core IPG motif of *m/z* 597 in abundance as well, corresponding to the Series A IPG, and to a lesser extent Series G and H IPG suggesting that, although we did not manage to detect the corresponding GIPCs by LC-MS, they were extracted. (Figs. 5D, E). These observations reveal that BCIN07g04350 is truly active on GIPCs originating from *A. thaliana* lipids and that through its activity, IPGs are released along with their corresponding phytoceramides.

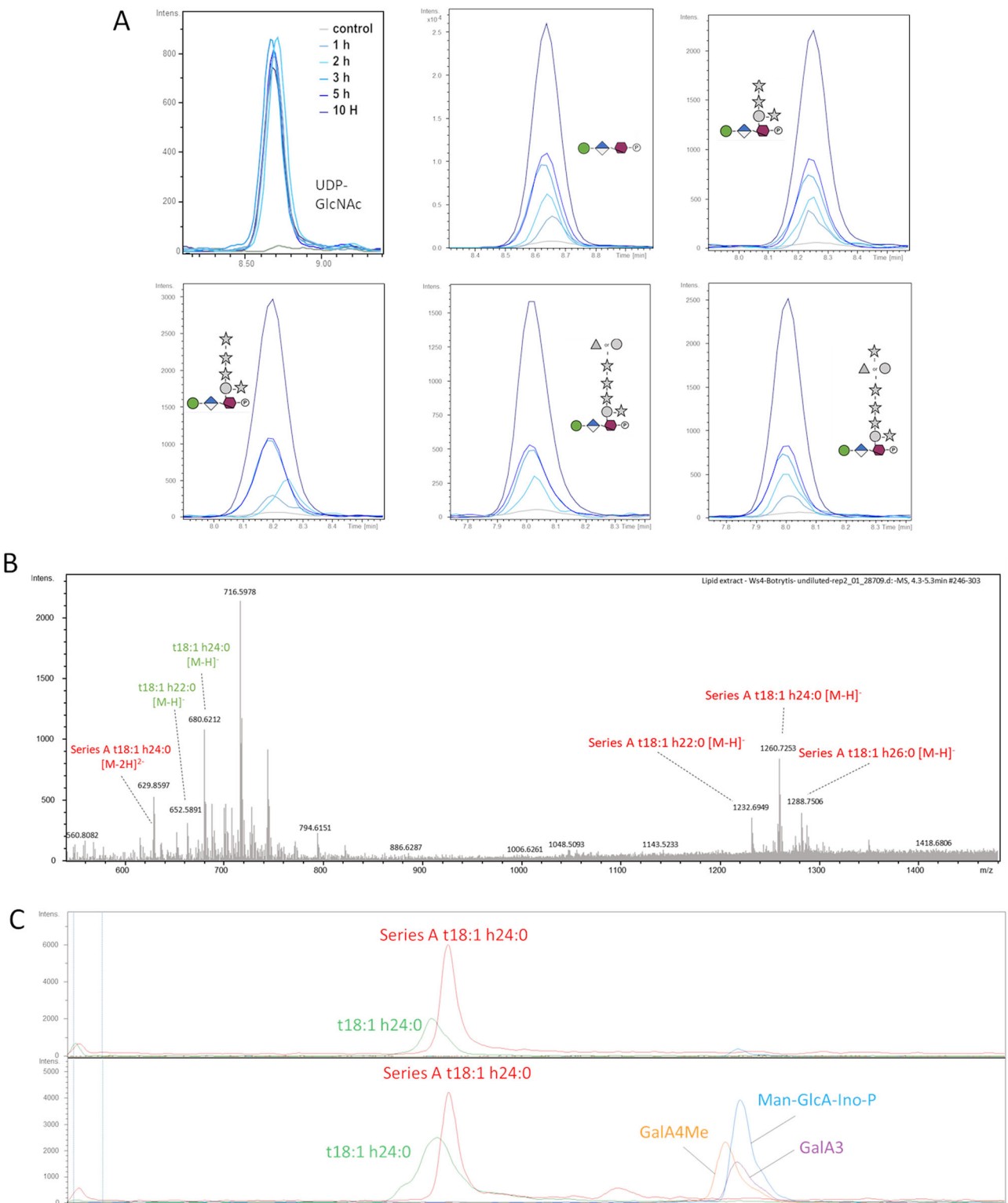

**Fig. 4 | *B. cinerea* exhibits GIPC phospholipase C activity. A** Extracted ion chromatograms of accumulated UDP-GlcNAc and IPGs over time. *B. cinerea* conidia were incubated with *A. thaliana* alcohol insoluble residues (AIR) and the production of each IPG was monitored using HPSEC-HRMS at 1, 2, 3, 5 and 10 h (light blue to dark blue). **B** Spectrum of GIPC-enriched extract acquired in the negative ion mode, prepared from *A. thaliana* AIR. **C** Upper panel: Extracted ion chromatograms of Series A GIPC t18:1 h22:0 (red) and its phytoceramide base t18:1 h22:0 (green) prepared from incubating *A. thaliana* AIR in a liquid culture medium not containing fungal conidia overnight. Lower panel: Extracted ion chromatograms of Series A GIPC t18:1 h22:0 (red), its phytoceramide base t18:1 h22:0 (green) and Man-GlcA-Ino-P (blue) IPG core obtained from incubating *A. thaliana* AIR with *B. cinerea* conidia in a liquid culture medium overnight. Presence of GalA3 (purple) and GalA4Me (yellow) suggests digestion of pectin is occurring.

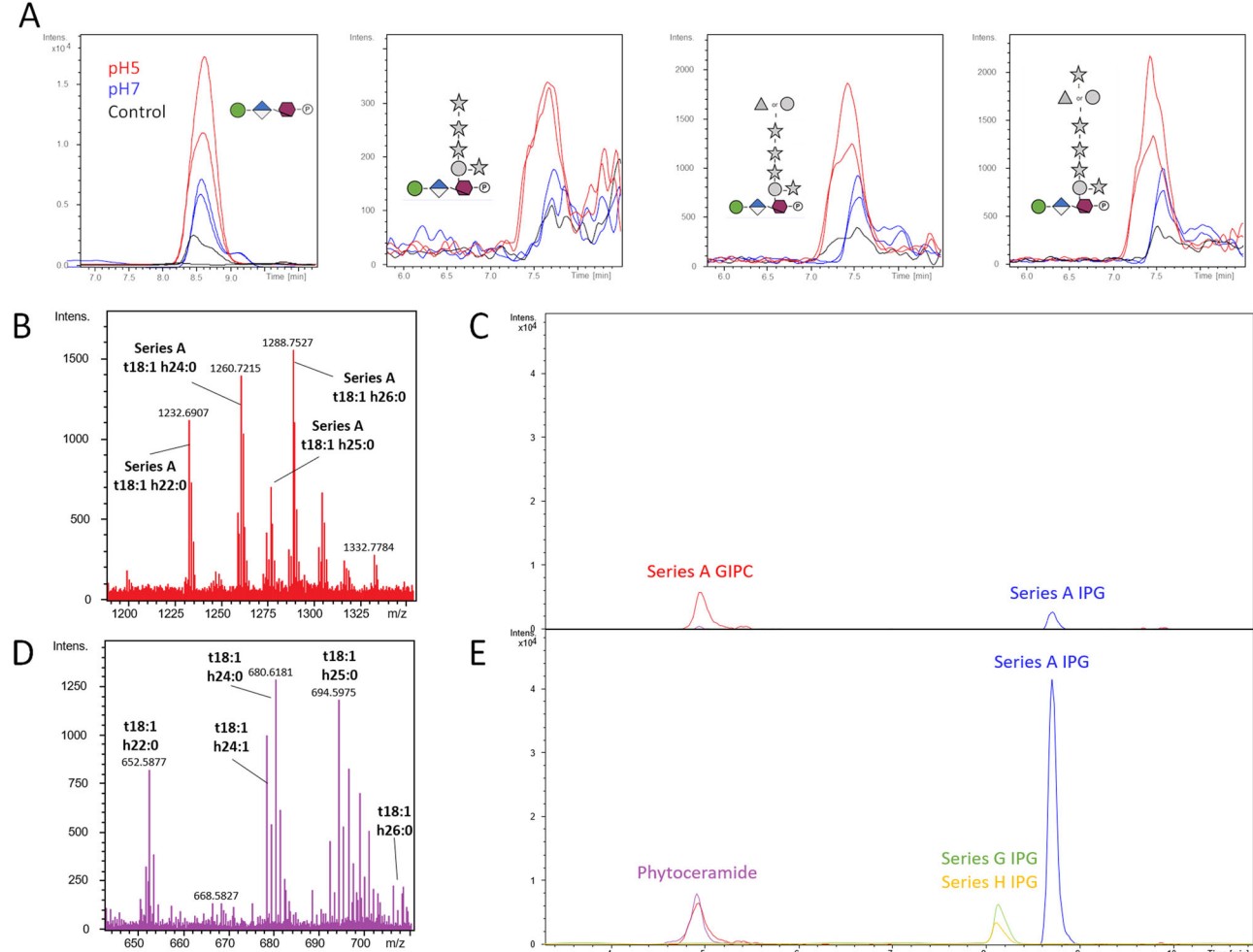

**Fig. 5 | *B. cinerea* is responsible for the production of Inosital Phosphate Glycans (IPGs) from plant membrane upon infection.** Extracted ion chromatograms of IPGs (**A**) produced by the putative GIPC-PLC1 encoded by *BCIN07g04350* gene from *A. thaliana* AIR at different pH values. **B** Spectrum of GIPC-enriched extract acquired in the negative ion mode. **C** Cumulative extracted ion chromatograms of most prominent phytoceramides (purple), GIPC series (red) and Series A (blue) IPG core present in the untampered GIPC-enriched lipid fraction prepared from *A. thaliana* AIR. **D** Spectrum of phytoceramides (purple) acquired in the negative ion mode. **E** Cumulative extracted ion chromatograms of most prominent phytoceramides (purple), GIPC series (red) and Series A (blue), Series G (green) and Series H (yellow) IPG present in the GIPC-enriched lipid fraction incubated with the putative GIPC-PLC1 encoded by the *BCIN07g04350* gene.

## IPG production correlates with lesion development

Last, since early putative *GIPC-PLC1* expression correlates with lesion size area (Table S1), we assessed if an increased virulence of *B. cinerea* would correlate with higher IPG production. We first incubated fresh *A. thaliana* leaves with solutions of conidia from BcWT, the *polygalacturonase1*[27] (*Bcpg1*) and *pectin lyase 1* (*Bcpnl1*) hypovirulent mutants affected in degradation of highly methylesterified pectins[28], the *pectin methylesterase 1/2* mutant (*Bcpme1/2*) which is slightly more virulent than BcWT despite its inability to degrade pectins efficiently[15] and *Bcacp1*, deficient in an acidic protease and moderately hypovirulent[29]. We next monitored IPG and xyloglucan-derived oligosaccharide production at 18 h after inoculation (Fig. 6). We observed that significantly fewer IPGs were accumulated when plants were infected with the *Bcpnl1* and *Bcpg1* hypovirulent strains and to a lesser extent with *Bcacp1* while no significant difference in IPG accumulation was observed in leaves infected by the *Bcpme1/2* strain (Fig. 6A). From this, it follows that IPG production is likely affected in low virulent strains, at least for the three mutants analyzed. Furthermore, IPG accumulation displays significant positive correlation with lesion development as described by the Pearson's rank correlation analysis between 72 hpi lesion area and IPG peak area at 18 hpi across the four *B. cinerea* genotypes, on the contrary to the xyloglucan fragment. Indeed, fewer xyloglucan fragments were found to be accumulating in plants infected by *Bcpme1/2* and *Bcpnl1* which display opposite virulence (Fig. 6B). This results rather underscore the fact that degradation of highly methylesterified pectins is needed prior to xyloglucan degradation.

## Discussion

We unveiled the presence of previously undiscovered oligosaccharides that accumulate during the interaction between plants and *B. cinerea*. Despite anticipating to characterize mainly pectin-derived oligosaccharides, our study revealed a prominent presence of xyloglucan fragments and more importantly, IPGs released upon infection from plasma membrane-bound GIPCs.

In plants, one IG has already been identified in plant culture cells[30,31]. However, to our knowledge, our study represents the first report of IPGs in plants, despite these compounds having been well-documented in mammals for over 30 years[32]. We demonstrated the presence of IPGs from Series A to H in cell wall extracts of uninfected plants, which are preferentially accumulated upon infection. The fact that *B. cinerea* was able to produce these IPGs from inert plant material prompted our investigations into the existence of genes encoding GIPC-cleaving enzymes in the *B. cinerea* genome. We successfully identified BCIN07G04350 as GIPC-phospholipase in *B. cinerea*, annotated as a sphingomyelinase, which is active on Series A to Series G and H GIPCs, indicating its potential role in the generation of these

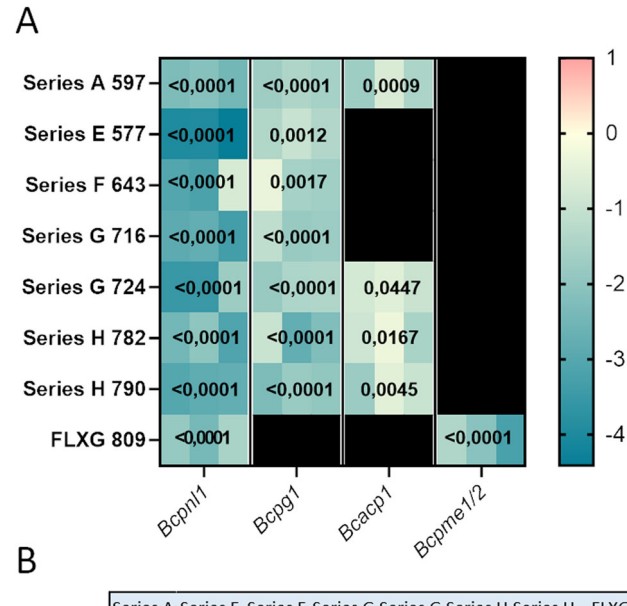

**Fig. 6 | IPG production is impaired in low-virulent *B. cinerea* mutant on the contrary to xyloglucan-derived OS. A** Heatmap displaying the log2 ratios of differentially accumulated IPGs during infection of *A. thaliana* with *Bcpnl1*, *Bcpg1*, *Bcacp1* and *Bcpme1/2* compared to BcWT. Each box indicates a single biological replication, *n* = 3. Statistical analysis was performed using one-way ANOVA analysis with Tukey's HSD test (p < 0.05). **B** Pearson's rank correlation coefficient between *A. thaliana* detached leaf lesion area and IPG peak area at 18 hpi across the four *B. cinerea* genotypes (*n* = 12).

IPGs during the infection interaction between the plant and *B. cinerea*. It has been previously shown these enzymes are present in the exo-proteome secreted during pathogenic interactions. By impairing *B. cinerea* secretory pathways through mutagenesis, the detectable quantity of this enzyme among 32 other carbohydrate-active enzymes drops, which in turn lowers the pathogenic potential of the fungus itself[26]. Moreover, exo-proteome and transcriptome analyses performed on four non-pathogenic *B. cinerea* transformants (*NPT1/4*), almost unaffected in hyphal growth but exhibiting defects in infection cushion formation and medium acidification, showed that *BcGIPC-PLC1* was down-accumulated in the four *NPT* together with many CAZy proteins known to play a major role in the degradation of the plant cell wall[33]. This highlights the place of GIPC-phospholipases in the carbohydrate-targeting secretome of *B. cinerea*.

It is worth noting that the non-specific phospholipase C4 (NPC4) in *A. thaliana* has been shown to be involved in GIPC hydrolysis in response to phosphate deficiency[34]. This is consistent with the fact that we were able to detect IPGs produced in non-infected leaves. Currently, the involvement of NPC4 and/or its close orthologs in generating IPGs together with BcGIPC-PLC1 upon infection cannot be ruled out.

Also, the accumulation of series A inositol glycan (*m/z* 517) suggests that a GIPC phospholipase D, which cleaves after the phosphate moiety, could be active as well. Phospholipase D activity has been detected in cabbage and in *A. thaliana*[34,35]. This GIPC-phospholipase prefers GIPCs containing two sugars as substrates[36] which aligns with the fact that Series A is the most produced IPG in our condition. However, we cannot rule out another scenario that would include a phospholipase C activity and a subsequent phosphorylase activity during infection.

Furthermore, the absence of series E to H IPGs in the *gmt1.3* mutant, which is deficient in adding mannose residues, suggests that mannose addition is indispensable before transfer of other sugars. Notably, this finding raises questions about the role of the absence of complex GIPCs in the *gmt1.3* mutant's severe phenotype and constitutive hypersensitive response[23]. It warrants further investigation before light can be shed on the contribution of IGs and IPGs to the phenotypic severity observed in the *gmt1.3* mutant. I(P)Gs could indeed potentially play a role in developmental process, hypothetically acting more broadly, perhaps as stress-regulators or signaling molecules. It has been shown that Series A IGs accumulate in cultured rose cells predominantly during the period of rapid cell growth[30], suggesting that their production does not inhibit cell division and might be involved in cell growth or be an indicator of plasma membrane remodeling. Interestingly, a glucuronyl-mannose disaccharide of unknown origin, which could be produced through IG or IPG hydrolysis, has been very recently shown to accumulate in salt resilient *A. thaliana* plants native to Cape Verde Islands, mutated for an alpha glycosidase encoding gene. It suggests that this type of molecule likely protect plant from salt stress, improving their root growth and hydric status[37]. Further research is needed to conclude whether IPGs are biologically active and play a role in plant defense regulation.

Finally, findings presented in this paper offer a glimpse into the dynamic interactions between the necrotrophic pathogen *B. cinerea* and its plant hosts while also broadening our understanding of enzymes and molecules released upon it. The involvement of lipids in plant–pathogen interaction and signal transduction has been described for many decades[38]. Some phytopathogen phospholipases have notably been characterized, such as PLA2 from *Verticillium longisporum* that targets host nuclear envelope phospholipids[39] and the phosphatidylinositol-specific phospholipases-C from *B. cinerea* and *Ceratocystis cacaofunesta*. However, to our knowledge, none of them targets GIPCs[40,41]. According to the phylogenetic analysis, GIPC-PLC activity might be widespread conserved among major fungal phytopathogen lineages, independently of their lifestyle, with a notable exception in the genus Alternaria that suggests potential functional divergence or alternative pathogenic mechanisms in these fungi.

Last, in addition to phospholipases that have been identified as essential virulence factors in several human-pathogenic fungi[42,43], bacterial sphingomyelinases, which enzymatically break down sphingomyelin in mammals (analogous to GIPCs in plants) release products identical to those generated by the host eukaryotic enzymes. These products aid in evading the host's immune response[44]. Whether the membrane degradation itself and/or the OS-like released upon this degradation are biologically active and play a role in the plant/pathogen interaction still needs to be deciphered.

## Materials and methods
### Plant Material and growth
Infection assays were performed on *A. thaliana* WT Wassilewskija (Ws-4) plants grown in soil in a growth chamber at 22 °C, 70% humidity, under irradiance of 100 mol.m$^{-2}$.s$^{-1}$ with a photoperiod of 8 h light/16 h dark. Seeds from *gmt1.3* mutants were a kind gift of J. Mortimer. *A. thaliana gmt1.3* and Colombia seeds were grown on 1/2 × MS media plates positioned vertically for 14 days under constant light at 23 °C. The tomato *Solanum lycopersicum* was cultivated in greenhouse conditions in soil.

The *B. cinerea* B05.10 collected from *Vitis* in Germany[45] was used as the wild-type (WT) reference strain. It was grown on potato dextrose agar at 23 °C under continuous light.

### Alcohol insoluble residue and cell wall water extract
Leaves and plantlets were grinded in five volumes of ethanol 96% and the supernatant was removed after centrifugation at 10,000 × g for 5 min. The pellet was washed with 70% ethanol with subsequent centrifugation steps until becoming discolored. Then it was dried in a speed vacuum concentrator at room temperature. Two volumes of distilled water were added to the alcohol insoluble residue obtained and left one hour at room temperature. The supernatant was again collected after centrifugation at 10,000 × g for 5 min and dried in a speed vacuum concentrator at room temperature.

## GIPC-enriched lipid fraction extraction

AIR was extracted using ethanol 70%, HCl 0.1 N at 70°C during 15 min. The GIPC-containing supernatant was collected and precipitated overnight at −20 °C. The GIPC-enriched pellet was then collected following centrifugation at 5000 rpm at 4 °C during 10 min. This pellet was resuspended in water.

## Fungal strains and growth

The wild-type *B. cinerea* B05.10 strain and Bc*pg1*[27] and Bc*pme1/2*[15] mutant strains were grown on potato dextrose agar (*Difco*[TM]) at 23 °C under continuous light. After 10 days, each strain produced a dense carpet of conidia.

## Oligosaccharides accumulated upon *B. cinerea* infection of *A. thaliana* and *S. lycopersicum* leaves

Oligosaccharides were produced and analyzed according to Voxeur et al.[15].

Briefly, after 10 days, the conidia of *B.cinerea* were washed from the surface of the plate using Gamborg's B5 basal medium, 2% (w/v) fructose and 10 mM phosphate buffer. Fungal hyphae were removed by filtering through a nylon mesh with 0.25 µL pore size. The concentration of conidia was determined using a Malassez cell and adjusted to a final concentration of $3.10^5$ conidia/mL. Isolated tomato leaves of 4-week-old plants or *A. thaliana* leaves of 5-week-old plants were immersed in a *B. cinerea* suspension (6 leaves for 10 ml of suspension at $3 \times 10^5$ conidia/ml) and incubated on a rotary shaker at 100 rpm at 23 °C during 12, 15 and 18 h. The liquid medium was collected at the three time-points and an equal volume of 96% ethanol was added for preservation. After centrifugation at $5000 \times g$ during 10 min, the supernatant was collected and dried in a speed vacuum concentrator at room temperature. The obtained pellet was then diluted. The equivalent of the digestate of 3 leaves of 5-week-old *A. thaliana* plants and similar leave surface of tomato was dried and diluted in 200 µl and of that, 10 µl were injected for MS analysis.

## Oligosaccharide and glycolipid analysis

Samples were diluted at 1 mg/ml in ammonium formate 50 mM, formic acid 0.1%. Chromatographic separation on high-performance size-exclusion chromatography was performed on an ACQUITY UPLC Protein BEH SEC Column (125 Å, 1.7 µm, 4.6 mm × 300 mm, Waters Corporation, Milford, MA, USA). Elution was performed in 50 mM ammonium formate, formic acid 0.1% at a flow rate of 400 µl/min and a column oven temperature of 40 °C. The injection volume was set to 10 µl. MS-detection was performed on a Bruker impact II QTOF in negative mode with the end plate offset set voltage to 500 V, capillary voltage to 4000 V, Nebulizer 40 psi, dry gas 8 l/min and dry temperature 180 °C. Major peaks were annotated following accurate mass annotation, isotopic pattern and MS/MS analysis. The MS fragmentation pattern is indicated according to the nomenclature of Domon and Costello[46]. For the targeted analysis, the theoretical exact masses were used with 4 significant figures with a scan width of 5 ppm. The resulting extracted ion chromatograms were integrated.

## Data processing

The d data files (Bruker Daltonics, Bremen, Germany) were converted to mzXML format using the MSConvert software (ProteoWizard package 3.0[47]) mzXML data processing, mass detection, chromatogram building, deconvolution, sample alignment, and data export were performed using MZmine 2.52 software (https://mzio.io/#mzmine). Mass list were built using a retention time window of 6.0–8.0 min and next, we used the ADAP chromatogram builder[48] with group size of scan 5, peak detection threshold of 800, a minimum highest intensity of 1500 and *m/z* tolerance of 0.01 m/z. We deconvoluted the data with the ADAP wavelets algorithm using the following setting: S/N threshold 8, peak duration range = 0.01–0.2 min RT wavelet range 0.02–0.1 min. Finally, we selected candidates with a surface area superior to 18,000 at 18 h post-inoculation and results were deisotoped manually.

## *BcGIPC-PLC* cloning and enzymatic activities

The CDS of the putative sphingomyelinase *BCIN07g04350* from *B. cinerea* (21-630 aa) was cloned in frame with the α-factor sequence allowing the protein secretion and the C terminal peptide containing the *c-myc* and 6XHis- epitopes into the vector pPICZαA (Proteogenix, https://www.proteogenix.science/fr). Codon optimization was used to improve the expression of the protein in *Pichia pastoris X33*. The yeast strain was transformed with 10 µg of the linearized plasmid digested by *SacI* enzyme according to the EasySelect[TM] yeast expression kit (Invitrogen). Transformants were isolated and analyzed for the presence of the insert using *AOX1* primers according to the EasySelect[TM] yeast expression kit (Invitrogen, 5′ *AOX1* primer: 5′-GAC TGG TTC AAA TTG ACA AGC-3′ and 3′ *AOX1* primer: 5′-GCA AAT GGC ATT CTG ACA TCC-3′). The successfully obtained transformants were grown in baffled flasks in 1.5 mL of buffered glycerol-complex medium, overnight at 30 °C using 100 mg/ml Zeocin. Cells were then collected by centrifugation and resuspended to an OD600 of 1.0 in 100 mL of buffered methanol complex medium. A final concentration of 0.5% (v/v) methanol was added every 24 h to maintain induction. After 72 h of induction, the culture was centrifuged at $1500 \times g$ for 10 min and the culture supernatant was loaded on Amicon[TM] ultra—4 ml–30 kDa cutoff (Merck millipore). To identify the recombinant protein present in the supernatant by Western blot, SDS-PAGE was transferred from resolving gel to PVDF blotting membrane using the appropriate cathode and anode buffers and a Trans-Blot TURBO Transfer System (Bio-Rad, Cat. No. 170-4155) at 120 V for 60 min. TBS-T (0.5% Tween 20 in TBS) was used as washing buffer and 5% non-fat dried milk in TBS-T was used as blocking reagent. Transferred proteins were incubated for 1 h at room temperature under shaking with 1:3000 dilution of anti-his antibody coupled with peroxidase (Sigma, Cat. No. A7058). After washes, the ECL kit (Cytiva) was used to detect the protein of interest according to the supplier's instructions. 200 µl of the concentrated supernatants of induced and non-induced cell culture were next incubated with 10 mg of alcohol insoluble residues obtained from *A. thaliana* 5-week-old plant and incubated overnight at 37 °C at pH5 or 7.

One of the transformants mentioned previously was used for a second round of transgene expression for the purpose of obtaining the putative sphingomyelinase *BCIN07g04350* at a higher concentration. The transformant was inoculated into 50 mL of buffered glycerol-complex medium containing 100 mg/ml of Zeocin[TM] (Thermo Ficher Scientific) in an Erlenmeyer flask of a 500 mL volume and incubated overnight at 30 °C. Cells were then collected by centrifugation and resuspended to an OD600 of 1.0 in 100 mL of buffered methanol complex medium and in a 1 L Erlenmeyer flask. Methanol was added to a final concentration of 0.5% (v/v) every 24 h to maintain induction of the transgene of interest. After 72 h of such a treatment, the culture was centrifuged at $1500 \times g$ for 10 min and supernatant containing the protein of interest was collected for further purification. A mock control sample was prepared in the same manner but the incubation in buffered methanol complex medium was exchanged with incubation in buffered glycerol complex medium and no methanol was added during the induction phase of 72 h. This culture was treated the same in the following steps. First, the cell culture supernatant was filtered manually with a Acrodisc® Syringe Filter 0.45 µm Supor® Membrane (PALL Life Sciences). Next, the filtered supernatant was used for immobilized metal ion affinity chromatography (IMAC). This was executed on the HisTrap[TM] (1 mL) excel column (cytiva), a commercially available column filled with separation resin precharged with nickel ions. The mentioned column was connected to a peristaltic pump that was used to feed it with a set flow rate of 1 mL/min. Following manufacturer's recommendations, the column was washed with 5 volumes of distilled water and then with 5 volumes of equilibration buffer (20 mM sodium phosphate, 0.5 M sodium chloride, pH 7.4). The total volume of the collected supernatant was loaded on the column next. The column was then washed with 20 volumes of wash buffer (20 mM sodium phosphate, 0.5 M sodium chloride, 10 mM imidazole, pH

7.4) and the enzyme of interest was eluted with 5 volumes of elution buffer (20 mM sodium phosphate, 0.5 M sodium chloride, 500 mM imidazole, pH 7.4). This substantially smaller volume was then applied on the PD SpinTrap G-25 single-use column (Cytiva) to remove eventual inorganic salts that could act inhibitory. The obtained fraction was further concentrated on the Amicon™ ultra—4 ml—10 kDa cutoff (Merck millipore). The obtained concentrates of the induced and the uninduced cell culture were then incubated with 10 mg of alcohol insoluble residues obtained from *A. thaliana* 5-week-old plant and incubated overnight at 37 °C at pH 5 each.

### Statistics and reproducibility
The number (n) of samples for each value is indicated in figures or figure legends. Significant differences for multiple comparisons were determined by two-way ANOVA with Tukey's HSD test ($p < 0.05$) as indicated in figure legends. Each experiment was repeated independently at least two times with consistent results.

### Reporting summary
Further information on research design is available in the Nature Portfolio Reporting Summary linked to this article.

### Data availability
All data are available in the main text or the Supplementary Data 1. Materials are available from the corresponding authors on reasonable request.

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

## Acknowledgements

J. Mortimer, M-C Soulié are thanked for providing *gmt1* seeds and B05.10 strain, respectively. This work has benefited from the support of IJPB's Plant Observatory technological platforms and from the following fundings : Université Paris Saclay Prematuration CDE2018-002330-IRE 2018-0024 OGome, French National Research Agency ANR-14-CE34-0010-03-PEC-TOSIGN and ANR-22-CE43-0013-WALLDERIVE and National Research Institute for Agriculture, Food and Environment tenure track grant.

## Author contributions

Conceptualization: A.V., S.V. Methodology: A.V., S.V., T.D., L.L. Investigation: A.V., L.L., J.R., J.S., A.F., A.D. Funding acquisition: A.V., S.V. Project administration: S.V. Supervision: A.V., S.V. Writing—original draft: A.V. Writing—review & editing: A.V., S.V., J.S., L.L., A.D.

## Competing interests

The authors declare no competing interests.

## Additional information

**Peer review information** *Communications Biology* thanks Giulia De Lorenzo, and the other, anonymous, reviewer for their contribution to the peer review of this work. Primary Handling Editor: David Favero. A peer review file is available. This manuscript has been previously reviewed at another Nature Portfolio journal. This document only contains reviewer comments and rebuttal letters for versions considered at *Communications Biology*.

