## [Peer Review file · Communications Biology]

A fungal phospholipase C involved in the degradation of plant glycosylinositol phosphorylceramides during Arabidopsis/Botrytis interaction

Corresponding Author: Dr Aline Voxeur

Version 0:

Reviewer comments:

Reviewer #1

(Remarks to the Author)

This study reports the discovery that inositol phosphate glycans (IPGs), not of fungal origin, are abundantly produced during the interaction between *Arabidopsis thaliana* and *Botrytis cinerea*, suggesting that the fungus targets the plant plasma membrane for degradation. IPGs can be released from plasma membrane-bound glycosylinositol phosphorylceramides (GIPCs), but their role in plants is still unknown.

The authors demonstrate the presence of IPGs from Series A to H in cell wall extracts of uninfected plants. They also conclude that plant enzymes are unlikely to be involved in IPG release during the interaction with *B. cinerea*, as their transcripts do not significantly vary during infection with hypo- or hypervirulent strains of the fungus. Instead, *B. cinerea* can degrade plant GIPCs to produce IPGs, as shown by the fungus's ability to release these IPGs from inert plant material of wild-type plants. Moreover, fewer IPGs accumulated when plants were infected with the Bcpg1 hypovirulent strain, suggesting a correlation between IPG release and fungal colonization ability.

In this work, a *B. cinerea* sphingomyelinase (BCIN07g4350) is identified and expressed in *Pichia pastoris*. The enzyme is active on Series A to H GIPCs, indicating its potential role in IPG generation during plant-fungal interactions.

The methods used are sound and appropriate (advanced biochemical methods, high-resolution mass spectrometry (HRMS), and heterologous expression). However, although the work is well performed and uncovers novel oligosaccharides, it is largely descriptive and technical. Including some details of the MS identification in the figure legends could facilitate reading. The study does not investigate the biological function of IPGs in plant defense responses or the role of the BCIN07g4350 enzyme in pathogenesis. Moreover, the evidence for the lack of involvement of plant phospholipases is limited to gene expression regulation, and no information is provided on the expression behavior of sphingomyelinase in *B. cinerea*.

SPECIFIC COMMENTS

TITLE: In the title "Plant glycosylinositol phosphorylceramides are degraded by a fungal phospholipase C during Arabidopsis/Botrytis interaction," the authors clearly attribute the generation of IPGs to a fungal phospholipase. However, in the abstract, they mention a "putative phospholipase C," and the manuscript does not demonstrate that the identified enzyme is indeed responsible for IPG release during the *B. cinerea*-plant interaction.

ABSTRACT, lines 20-23: The authors state: "Contrary to the conventional belief that *B. cinerea* releases solely cell wall degrading enzymes, this research suggests that *B. cinerea* enzymatic machinery also aims at degrading the plant plasma membrane." While it is understandable to present results as a novel paradigm, it is an overgeneralization to say "the conventional belief that *B. cinerea* releases solely cell wall degrading enzymes." Do the authors really think the community believes this? Likely, the authors mean that the role of enzymes other than CWDEs in *B. cinerea* pathogenesis is poorly characterized. Additionally, the manuscript lacks citations on phospholipases in phytopathogenic fungi (e.g., 10.3390/plants10061098, 10.1111/mpp.13352, 10.1186/s12864-018-4440-4, 10.1111/j.1365-2958.2008.06105.x) and evidence that phospholipases are essential virulence factors in several human-pathogenic fungi, including *Candida albicans* (41), *Cryptococcus neoformans* (12), and *Aspergillus fumigatus* (10.1016/j.plipres.2015.11.003, 10.1128/IAI.05830-11).

Page 1 lines 36-37. Oligosaccharins "are known for their diverse effects in plants, including promoting growth in low- and

normal-light conditions (3-7) and anti-auxin activity (8,9)." The references to the literature are poor. The authors should cite the most detailed studies on the anti-auxin activity of oligosaccharins, specifically oligogalacturonides.

Page 1 lines 37-39. « Moreover, while we know for decades that oligogalacturonides (OGs), α -1,4-linked galacturonic acid oligomers deriving from pectins, trigger plant defense responses upon exogenous application". The authors neglect that in muro release of OGs has been achieved by inducible expression of the so-called OG-machine, triggering defense responses. They should reference this.

Page 1 lines 39-40: The term "oligosaccharins" is misplaced here. Oligosaccharins are oligosaccharides with biological activity. The authors should specify which novel oligosaccharides with biological activity they have identified, or otherwise use the term "oligosaccharides."

Page 2, line 1: The authors should clarify the purpose of this study. Was it conceived as an advancement of the previous study (<https://doi.org/10.1073/pnas.1900317116>)? What are the main differences (methodological, experimental setup, kinetic study of early infection stages, deeper characterization)?

Page 2, line 1. The authors should clarify why this study was conceived. Was for as an advance of the previous study (<https://doi.org/10.1073/pnas.1900317116>)? What are the main differences (methodological? Experimental set up? Kinetic study of early stages of infection? Deeper characterization?).

Page 2, lines 13-17: "This analysis led to the identification of 36 candidates, which included 16 candidates with predicted formulae and retention time 15 compatible with OGs we have previously identified (13), four of which are likely oxidized through activity of plant OG-oxidizing enzymes from the berberine bridge enzyme-like family (14) while 12 were unsaturated (-H₂O) and likely result from pectin lyase activity (13) (Fig. 1B). Why were only short oxidized OGs identified? Could this be due to technical issues?

Page 6 lines 9-10. "Also, the ions we observed at m/z 435 (C₁₂H₂₀O₁₂P) and 355 (C₁₂H₂₀O₁₂) were absent in the WT samples (Fig. 3A and B)".
Rephrase as follows: Also, we observed ions at m/z 435 (C₁₂H₂₀O₁₂P) and 355 (C₁₂H₂₀O₁₂) that were absent in the WT samples (Fig. 3A and B)

Page 6, line 21. To further confirm...

Page 7, line 17 "overexpressed". Better "upregulated"

Page 7, line 18" observed that none of them were down- or up in" Better "we observed that none of them were differentially expressed (neither downregulated nor upregulated) in..."

Page 8 lines "IPG production can be considered as correlated with fungal virulence, at least for the two mutants analyzed". Am I missing something? Only one mutant showed evidence of correlation....

Page 8, line 28-36. For completeness, the experiment should include AIR prepared with hot ethanol.

Page 10, lines 19-21 and 25-30: "For the purpose of investigating *B. cinerea* GIPC phospholipases, we selected from published transcriptome profiles (22) the most abundant transcript encoding a putative sphingomyelinase in germinating *B. cinerea* conidia (BcGIPC-PLC1)". " also poses the question if *B. cinerea* might secrete other GIPC-PLCs with varying substrate specificities to degrade highly glycosylated GIPCs." The authors should explain how many putative sphingomyelinases are produced by *B. cinerea* in germinating conidia and their relative transcript levels according to transcriptome data. Additionally, they should show how these enzymes are expressed when *B. cinerea* grows on AIR substrates.

Reviewer #2

(Remarks to the Author)

-Figure 3E: The authors should also show phospholipase D cleavage sites.

-Figure 4A and 4B: The authors used Bcpg1 and Bcpme1/2 in their analysis, but are these meaningful? Figure 4A shows the expression of lipase genes in *Arabidopsis thaliana*. However, the expression of these lipase genes does not necessarily increase or decrease during infection, and changes in enzyme activation or localization may occur. Without mentioning them, the contribution of *Arabidopsis* endogenous lipase during infection cannot be ruled out. In addition, in Figure 4B, the amount of IPG is indeed decreased during Bcpg1 infection, but is unchanged during Bcpme1/2 infection compared to BcWT. However, the authors described that "plasma membrane degradation is linked to infection severity" and "IPG production can be considered as correlated with fungal virulence". These are incorrect. Are the gene expression, protein levels, or secretion of BCIN07g4350 in Bcpg1 and Bcpme1/2 during infection different from BcWT? If the authors present data using Bcpg1 and Bcpme1/2, the authors should clarify the meaning of their analysis.

-Figure S3A and S3B: The authors write that the purpose of Figure 3B is "To further confirm that IPGs are not of fungal origin". If this is the case, it should be placed in the section "Inositol phosphate glycans originate from plant sphingolipids". The authors should scrutinize the purpose and content of the study. Overall, the section "B. cinerea secretes phospholipases responsible for the production of Inositol Phosphate Glycans" is difficult to read and understand. Overall, a rewrite is required.

-From Figures 4C, 4E, 4F, and 5, it is clear that phospholipase C derived from *B. cinerea* is a major contributor to IPG production during infection. However, as pointed out above, there are currently few data to deny that Arabidopsis endogenous enzymes act during infection. Therefore, the authors should consider the preferential involvement of *B. cinerea* phospholipase C in IPG production during infection, without denying the function of endogenous PLCs in Arabidopsis.

-Discussion: The main points of this manuscript are the detection of IPGs and IGs during infection and the identification of novel PLC, but the discussion is very limited. I would like more in-depth discussion on the function of IPG generated during infection and the characterization of the newly identified PLC (e.g., whether it exists in closely related species).

Version 1:

Reviewer comments:

Reviewer #1

(Remarks to the Author)

I would change the title in "A fungal phospholipase C in the degradation of plant glycosylinositol phosphorylceramides during Arabidopsis/Botrytis interaction".

All the criticisms have been addressed.

Reviewer #2

(Remarks to the Author)

Minor points

-In the bottom in Figure 3, the tops of Pentose and Hexose symbols are missing.

-In Figure 5E, the upper part of the peak of phytoceramides is missing.

-(P8, L16) Series A GIPCs composed of long-chain base "and" t18:1 and fatty acid chains h22:0 to h26:0
Is "and" necessary?

First, we would like to thank the reviewers for their thorough and insightful reviews of our manuscript and the time and effort they have invested in this process. Their valuable feedback and constructive comments have greatly contributed to enhancing the clarity and quality of our work. We have carefully addressed each of their concerns point by point.

Reviewer #1 (Remarks to the Author):

SPECIFIC COMMENTS

TITLE: In the title "Plant glycosylinositol phosphorylceramides are degraded by a fungal phospholipase C during Arabidopsis/Botrytis interaction," the authors clearly attribute the generation of IPGs to a fungal phospholipase. However, in the abstract, they mention a "putative phospholipase C," and the manuscript does not demonstrate that the identified enzyme is indeed responsible for IPG release during the *B. cinerea*-plant interaction.

The title has been changed accordingly.

ABSTRACT, lines 20-23: The authors state: "Contrary to the conventional belief that *B. cinerea* releases solely cell wall degrading enzymes, this research suggests that *B. cinerea* enzymatic machinery also aims at degrading the plant plasma membrane". " While it is understandable to present results as a novel paradigm, it is an overgeneralization to say "the conventional belief that *B. cinerea* releases solely cell wall degrading enzymes." Do the authors really think the community believes this? Likely, the authors mean that the role of enzymes other than CWDEs in *B. cinerea* pathogenesis is poorly characterized.

We agree that the original phrasing could be seen as an overgeneralization. To address this, we have modified the sentence to better reflect our findings and avoid overstatement. The revised sentence now reads: "In addition to the well-characterized role of *B. cinerea* in releasing cell wall-degrading enzymes, this research suggests that *B. cinerea*'s enzymatic machinery may also target the degradation of the plant plasma membrane." This revision more accurately conveys that while cell wall-degrading enzymes are well-studied, the role of other enzymes in *B. cinerea* pathogenesis is less understood.

Additionally, the manuscript lacks citations on phospholipases in phytopathogenic fungi (e.g., 10.3390/plants10061098, 10.1111/mpp.13352, 10.1186/s12864-018-4440-4, 10.1111/j.1365-2958.2008.06105.x) and evidence that phospholipases are essential virulence factors in several human-pathogenic fungi, including *Candida albicans* (41), *Cryptococcus neoformans* (12), and *Aspergillus fumigatus* (10.1016/j.plipres.2015.11.003, 10.1128/IAI.05830-11).

These references have been added in the discussion page 13 lines 43-46

Page 1 lines 36-37. Oligosaccharins “are known for their diverse effects in plants, including promoting growth in low- and normal-light conditions (3-7) and anti-auxin activity (8,9).” The references to the literature are poor. The authors should cite the most detailed studies on the anti-auxin activity of oligosaccharins, specifically oligogalacturonides.

Page 1, line 37, the following references have been added: [10.1104/pp.111.184663](https://doi.org/10.1104/pp.111.184663) ; [10.1111/j.1399-3054.1988.tb09157.x](https://doi.org/10.1111/j.1399-3054.1988.tb09157.x)

Page 1 lines 37-39. « Moreover, while we know for decades that oligogalacturonides (OGs), α -1,4-linked galacturonic acid oligomers deriving from pectins, trigger plant defense responses upon exogenous application”. The authors neglect that *in muro* release of OGs has been achieved by inducible expression of the so-called OG-machine, triggering defense responses. They should reference this.

Page 1, line 40, we added “In addition, *in muro* release of OGs has been achieved through expression of a fungal polygalacturonase gene (PG) fused with a gene encoding a plant polygalacturonase-inhibiting protein (16).”

Page 1 lines 39-40: The term “oligosaccharins” is misplaced here. Oligosaccharins are oligosaccharides with biological activity. The authors should specify which novel oligosaccharides with biological activity they have identified, or otherwise use the term “oligosaccharides.”

Done

Page 2, line 1: The authors should clarify the purpose of this study. Was it conceived as an advancement of the previous study (<https://doi.org/10.1073/pnas.1900317116>)? What are the main differences (methodological, experimental setup, kinetic study of early infection stages, deeper characterization)?

To clarify the purpose of this study, we have added the following sentence to page 2, line 3: “To achieve a comprehensive characterization of the oligosaccharides generated during the *A. thaliana*/*B. cinerea* interaction, beyond just pectin-derived oligosaccharides, we performed the structural characterization of the oligosaccharides accumulated over time.”

Page 2, lines 13-17: “This analysis led to the identification of 36 candidates, which included 16 candidates with predicted formulae and retention time 15 compatible with OGs we have previously identified (13), four of which are likely oxidized through activity of plant OG-oxidizing enzymes from the berberine bridge enzyme-like family (14) while 12 were unsaturated (-H₂O) and likely result from pectin lyase activity (13) (Fig. 1B). Why were only short oxidized OGs identified? Could this be due to technical issues?”

This cannot be due to technical issues as we efficiently detect non-oxidized OGs. Based on the literature, OGOx typically oxidizes long, fully demethylated OGs (Benedetti et al., 2008) However, we observed that these long OGs are rapidly degraded by *B. cinerea*'s enzymatic machinery (Voxeur et al., 2019). This rapid degradation likely explains why we only identified short oxidized OGs in our analysis. The results of our analysis are consistent with these observations, as the short oxidized OGs are the stable products detectable under the conditions used.

Page 6 lines 9-10. "Also, the ions we observed at m/z 435 (C₁₂H₂₀O₁₂P) and 355 (C₁₂H₂₀O₁₂) were absent in the WT samples (Fig. 3A and B)". Rephrase as follows: Also, we observed ions at m/z 435 (C₁₂H₂₀O₁₂P) and 355 (C₁₂H₂₀O₁₂) that were absent in the WT samples (Fig. 3A and B)
Done

Page 6, line 21. To further confirm...

Done

Page 7, line 17 "overexpressed". Better "upregulated"

Page 7, line 18 "observed that none of them were down- or up in" Better "we observed that none of them were differentially expressed (neither downregulated nor upregulated) in..."

This part has been deleted from the manuscript.

Page 8 lines "IPG production can be considered as correlated with fungal virulence, at least for the two mutants analyzed". Am I missing something? Only one mutant showed evidence of correlation....

We have added results obtained with the *Bcpnl1* (Davière et al., 2024) and *Bcacp1* (Soulié et al., 2020) mutants to provide additional evidence. This inclusion strengthens the correlation between IPG production and fungal virulence, as our analysis now includes data from multiple mutants. We have revised the text to accurately reflect these expanded findings and clarify the relationship between IPG production and fungal virulence. In addition, we moved this part page 11-12 and figure 6

Page 8, line 28-36. For completeness, the experiment should include AIR prepared with hot ethanol.

This has been added Fig. S4.

Page 10, lines 19-21 and 25-30: "For the purpose of investigating *B. cinerea* GIPC phospholipases, we selected from published transcriptome profiles (22) the most abundant transcript encoding a putative sphingomyelinase in germinating *B. cinerea*

conidia (BcGIPC-PLC1)". " also poses the question if *B. cinerea* might secrete other GIPC-PLCs with varying substrate specificities to degrade highly glycosylated GIPCs." The authors should explain how many putative sphingomyelinases are produced by *B. cinerea* in germinating conidia and their relative transcript levels according to transcriptome data.

Page 10, we have revised the relevant sections to address this concern. We now provide a detailed explanation of the number of putative sphingomyelinases produced by *B. cinerea* in germinating conidia, as well as their relative transcript levels based on transcriptome data. Additionally, we have incorporated more compelling data from the literature to support our focus on BcGIPC-PLC1, emphasizing its significance among the identified sphingomyelinases. This revised section clarifies our rationale for selecting BcGIPC-PLC1 for further investigation.

Additionally, they should show how these enzymes are expressed when *B. cinerea* grows on AIR substrates.

Page 10, lines 17-21, we have incorporated data from the literature to address this. Specifically, we added information showing that BcGIPC-PLC1 is expressed when the fungus is grown on cell-wall substrates. Additionally, data from Souibgui et al., 2021 (10.3389/fpls.2021.668937) indicate that this enzyme, unlike the two other putative sphingomyelinases, was secreted when conidia were grown in CCPX medium, which contains polysaccharides such as carboxymethyl cellulose, polygalacturonic acid, and xylan. This evidence supports the hypothesis that the expression of BCIN07g04350 may be upregulated in presence of *Arabidopsis thaliana* AIR and play a role in IPG production.

Reviewer #2 (Remarks to the Author):

-Figure 3E: The authors should also show phospholipase D cleavage sites.
Done

-Figure 4A and 4B: The authors used Bcpg1 and Bcpme1/2 in their analysis, but are these meaningful? Figure 4A shows the expression of lipase genes in *Arabidopsis thaliana*. However, the expression of these lipase genes does not necessarily increase or decrease during infection, and changes in enzyme activation or localization may occur. Without mentioning them, the contribution of *Arabidopsis* endogenous lipase during infection cannot be ruled out.

We have removed the data related to *Arabidopsis thaliana* lipase genes from Figure 4A and have revised the text to better reflect the limitations of our analysis. We acknowledge that

changes in enzyme activation or localization, which were not addressed in our initial presentation, could impact the role of Arabidopsis endogenous lipases during infection. We have adjusted our discussion to clarify these points and focus on the relevance of the *Bcpg1* and *Bcpme1/2* mutants in our analysis (see page 11)

In addition, in Figure 4B, the amount of IPG is indeed decreased during *Bcpg1* infection, but is unchanged during *Bcpme1/2* infection compared to BcWT. However, the authors described that "plasma membrane degradation is linked to infection severity" and "IPG production can be considered as correlated with fungal virulence". These are incorrect. Are the gene expression, protein levels, or secretion of BCIN07g4350 in *Bcpg1* and *Bcpme1/2* during infection different from BcWT? If the authors present data using *Bcpg1* and *Bcpme1/2*, the authors should clarify the meaning of their analysis.

To address these concerns, we have incorporated data from two additional *Botrytis* mutants into our analysis and calculated the Pearson rank correlation coefficient to better refine and contextualize our findings. This additional analysis helps clarify the relationship between IPG production and fungal virulence (Fig. 6).

-Figure S3A and S3B: The authors write that the purpose of Figure 3B is "To further confirm that IPGs are not of fungal origin". If this is the case, it should be placed in the section "Inositol phosphate glycans originate from plant sphingolipids".

We agreed that the introduction to this part was misleading and have since revised it for clarity.

The authors should scrutinize the purpose and content of the study. Overall, the section "B. cinerea secretes phospholipases responsible for the production of Inositol Phosphate Glycans" is difficult to read and understand. Overall, a rewrite is required.

Pag 7, we have revised the first part of this section.

-From Figures 4C, 4E, 4F, and 5, it is clear that phospholipase C derived from *B. cinerea* is a major contributor to IPG production during infection. However, as pointed out above, there are currently few data to deny that Arabidopsis endogenous enzymes act during infection. Therefore, the authors should consider the preferential involvement of *B. cinerea* phospholipase C in IPG production during infection, without denying the function of endogenous PLCs in Arabidopsis.

This has been modified throughout the text.

-Discussion: The main points of this manuscript are the detection of IPGs and IGs during infection and the identification of novel PLC, but the discussion is very limited. I would like more in-depth discussion on the function of IPG generated during infection and the characterization of the newly identified PLC (e.g., whether it exists in closely related species).

In response, we have added a phylogenetic analysis in the Results section to address the characterization of the newly identified PLCs, including their presence in closely related species (Fig. S5). Additionally, page 13, lines 30-38, we have expanded the Discussion section to provide a more detailed exploration of the putative functions of IPGs generated during infection. This includes a thorough examination of their potential roles and implications in the infection process.